# Realization of high-efficiency fluorescent organic light-emitting diodes with low driving voltage

Amin Salehi[1], Chen Dong[2], Dong-Hun Shin[3], Liping Zhu [2], Christopher Papa[4], Anh Thy Bui[4], Felix N. Castellano [4] & Franky So [2]

It is commonly accepted that a full bandgap voltage is required to achieving efficient electroluminescence (EL) in organic light-emitting diodes. In this work, we demonstrated organic molecules with a large singlet-triplet splitting can achieve efficient EL at voltages below the bandgap voltage. The EL originates from delayed fluorescence due to triplet fusion. Finally, in spite of a lower quantum efficiency, a blue fluorescent organic light-emitting diode having a power efficiency higher than some of the best thermally activated delayed fluorescent and phosphorescent blue organic light-emitting diodes is demonstrated. The current findings suggest that leveraging triplet fusion from purely organic molecules in organic light-emitting diode materials offers an alternative route to achieve stable and high efficiency blue organic light-emitting diodes.

[1] Department of Physics, North Carolina State University, Raleigh, NC 27607, USA. [2] Department of Material Science and Engineering, North Carolina State University, Raleigh, NC 27606, USA. [3] Department of Chemical and Biomolecular Engineering, North Carolina State University, Raleigh, NC 27606, USA. [4] Department of Chemistry, North Carolina State University, Raleigh, NC 27607, USA. Correspondence and requests for materials should be addressed to F.S. (email: fso@ncsu.edu)

To form excitons in organic molecules in an organic light-emitting diode (OLED) via electrical excitation, electrons and holes must be injected to their respective frontier orbitals, i.e., the lowest unoccupied molecular orbital (LUMO) and highest occupied molecular orbital (HOMO) of the emitting molecule in the emissive layer. Excitons are formed when injected electrons and holes form bound pairs having a Coulombic potential larger than the room-temperature thermal energy[1–3]. Upon their formation, triplet and singlet excitons have the same degenerate energetics which is equal to the HOMO-LUMO gap energy, and as electrons and holes approach each other, the triplet and the singlet energy levels separate. The magnitude of this singlet-triplet splitting is determined by the exchange energy or the HOMO-LUMO orbital overlap[4,5]. Although triplet excitons can have much lower energy compared to singlet excitons, to form either species, electrical energy equivalent to the HOMO-LUMO gap is required to turn on an OLED.

Triplet-triplet annihilation (TTA), also known as triplet fusion (TF), is a process in which two triplet excitons annihilate and produce a higher energy triplet or singlet exciton[6]. In fluorescent molecules, TF leads to the formation of singlet excitons and hence electro-fluorescence, also known as P-type delayed fluorescence, which is due to the much longer lifetime of triplet excitons compared to singlet excitons[7]. Initially it was postulated that annihilation of two triplet excitons would create a singlet exciton with a 25% probability and a triplet exciton with a 75% probability, following spin statistics. However, in systems where the energy of the first excited triplet state ($T_1$) is about half of the first excited singlet state ($S_1$), i.e., $2T_1 = S_1$, Kondakov et al. showed that triplets can generate singlets via TF with a conversion efficiency close to 100%[8,9]. Monkman et al. also observed that for efficient TF leading to singlet emission, the higher-order triplet states ($T_n$) must be larger than twice the first triplet state, i.e., $T_n > 2T_1$, to prevent TF from producing higher-order non-radiative triplet excitons instead of radiative singlet excitons[10]. One of the most well-known materials which satisfies the energetic requirements for obtaining 100% singlet yield from TF is rubrene, which is a tetracene-based molecule[8,11]. There are other organic molecules that show delayed fluorescence due to TF, with different singlet production efficiencies and rates[12,13]. In addition to yellow emitting tetracene-based molecules, blue-emitting anthracene-based molecules such as 9,10-diphenylanthracene (DPA) are known to have delayed fluorescence due to triplet fusion[9,14].

While it is commonly accepted that generation of singlet and triplet excitons in an OLED requires the full bandgap voltage, recently we along with others reported sub-bandgap EL in rubrene-based devices with triplet excitons generated by energy transfer from charge-transfer (CT) excitons, also known as "exciplex" excitons[15,16]. The reason for electroluminescence (EL) at sub-bandgap voltages was that the formation of the lower energy exciplex excitons does not require the full bandgap energy. Figure 1 presents a schematic diagram of the energy transfer and up-conversion emission mechanism in CT-assisted OLEDs with EL at sub-bandgap voltages. Although these CT-assisted devices can have EL at sub-bandgap voltages corresponding to the energy of the CT state, they are inherently inefficient due to the energy down-flow (quenching) from the emitter back to the CT-state, and therefore the EL quantum efficiency is typically low, and high efficiency sub-bandgap OLEDs remain out of reach[15–19]. In this work, we demonstrate that organic molecules with a large singlet-triplet splitting which also exhibit P-type delayed fluorescence can form triplet excitons via direct charge injection, and result in efficient EL at sub-bandgap voltages close to their triplet energy without the presence of any exciplex states. Finally, taking advantage of direct formation of triplets by charge injection, we demonstrate a high efficiency deep blue fluorescent OLED with

a sub-bandgap turn-on voltage of 2.4 V, having luminances of 100 cd/m$^2$ and 1000 cd/m$^2$ at 2.9 and 3.4 V, respectively, and an external quantum efficiency (EQE) close to 10% at 1000 cd/m$^2$. The resulting operating power efficiency (PE) of 14.5 lm/W at 1000 cd/m$^2$ is higher than that of some of the best blue phosphorescent OLEDs, making it quite promising as a stable blue OLED for display applications.

## Results

**Materials Selection**. To design an efficient TF blue fluorescent OLED, 4,4′-bis(9-ethyl-3-carbazovinylene)-1,1′-biphenyl (BCzVBi) is chosed as the emitter and 9-[4-(10-phenyl-9-anthryl) phenyl]-9H-carbazole (CzPA) is selected as the host. CzPA has been previously used as a host in efficient blue fluorescent OLEDs with a long operational lifetime[20]. It is a diphenylanthracene-based molecule with a carbazole side group. Diphenylanthracene-based molecules are well known for their triplet-fusion delayed fluorescence and the carbazole group facilitates the hole transport of the molecule[21]. We chose BCzVBi as the blue fluorescent dopant for two reasons. First, due to the larger triplet energy of BCzVBi (1.94 eV) compared to CzPA (1.77 eV), triplet excitons will diffuse to the host molecules and have a larger fusion probability due to the larger concentration of the host molecules. Second, the large overlap between BCzVBi's absorption and CzPA's emission spectra, as shown in Supplementary Fig. 1, leads to efficient Förster energy transfer from CzPA to BCzVBi.

**Single layer OLEDs**. To characterize the TF properties of the emitter and host materials, single layer OLEDs using un-doped BCzVBi and CzPA as the emitter were first fabricated. The single-layer OLEDs have the following structure: indium tin oxide (ITO) / PEDOT (30 nm) / emitter (40 nm) / Cs$_2$CO$_3$ (1 nm) / Aluminum (100 nm). In this work PEDOT refers to poly(3,4-ethylenedioxythiophene) polystyrene sulfonate (PEDOT:PSS) and is used as the hole injection layer (HIL). Here, an electron transport layer (ETL) was not used as electron injection is enabled directly through the cathode.

Unexpectedly, in the BCzVBi device, EL at applied voltages as low as 2.10 V was observed, with emitted photon energies up to 3.0 eV, well in excess of the bandgap. Figure 2 presents the device current density and the photocurrent measured as a function of the bias voltage. To measure the EL, two kinds of photodetectors were applied: a photomultiplier tube (PMT) detector for very low light intensities and a Si-photodiode for medium and higher light intensities. The sensitivity threshold of the photodiode is approximately 0.1 cd/m$^2$ and its corresponding photocurrent is marked on the luminance plot in Fig. 2a. Figure 2b shows the molecular structure of BCzVBi and the device structure of the single-layer device while Fig. 2c displays the EL spectrum. Since there is no hole transport layer (HTL) or ETL presented in the device, these results clearly show that triplet excitons are formed under low voltage.

In the context of TF photochemical up-conversion, it is well established that the up-converted emission intensity displays a quadratic to linear trend as a function of input light excitation intensity between low to high photon flux[22]. These observations are a result of traversing two kinetic limits for the sensitized TF process, the weak-annihilation and strong-annihilation regimes, which only become apparent from the time-integrated expressions[13,22]. In the weak-annihilation limit, the first-order decay (and pseudo-first-order quenching by O$_2$) of the triplet excited state dominates the kinetics resulting in the quadratic power dependence where the fluorescence emission is proportional to the square of the excited triplet concentration. When the TF process dominates, bimolecular triplet annihilation now

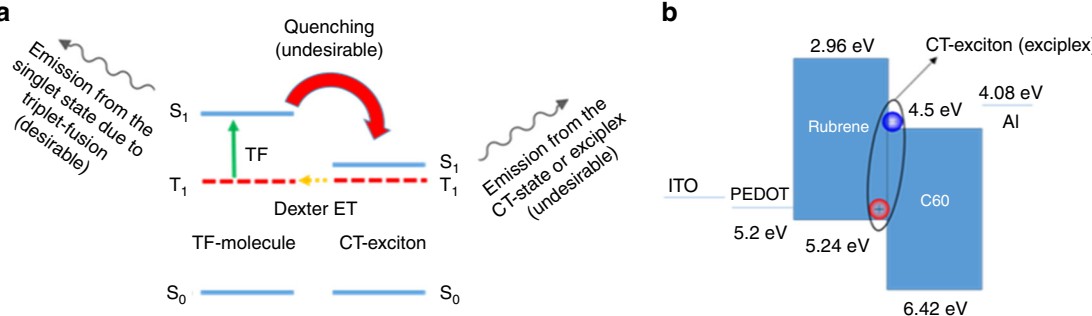

**Fig. 1** Working mechanism of CT-assisted OLEDs. **a** Energy transfer schematic and mechanism of previously reported OLEDs with EL at sub-bandgap voltages. **b** Energy level diagram of rubrene and $C_{60}$ based exciplex OLED

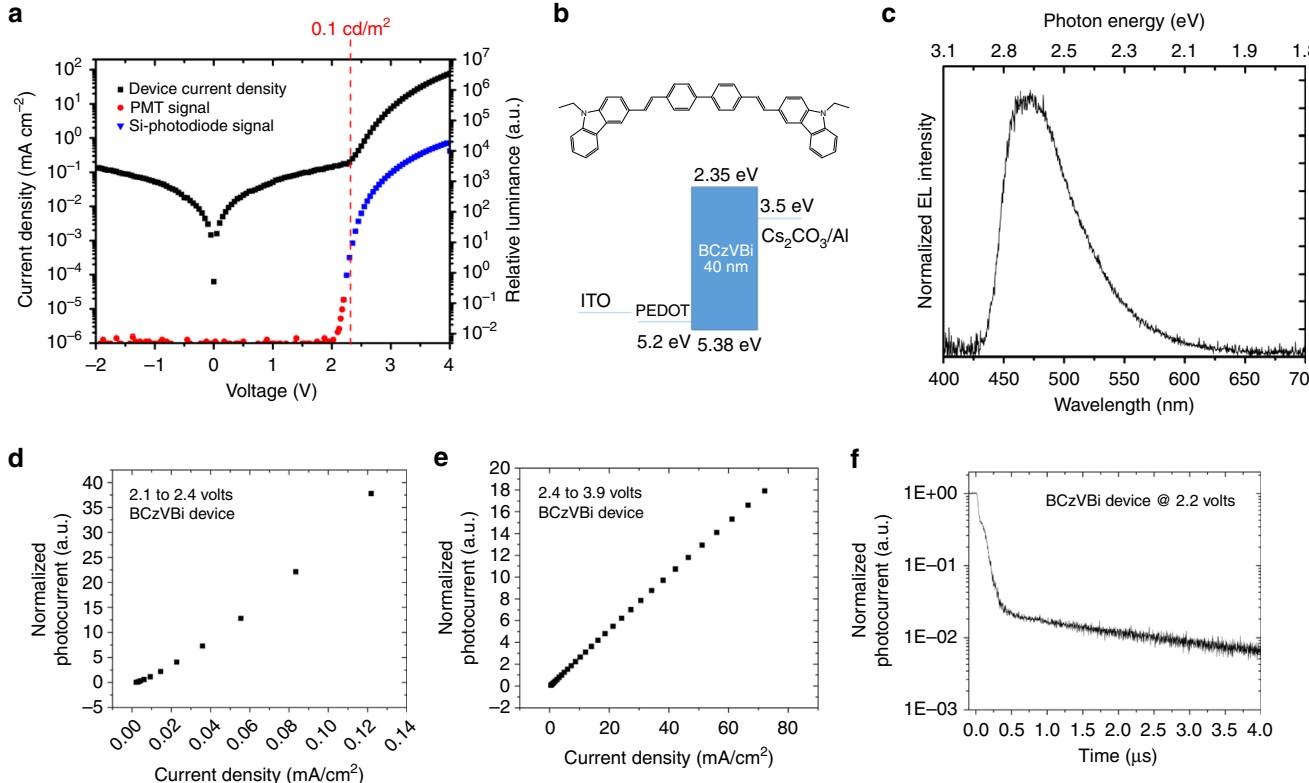

**Fig. 2** Single layer BCzVBi device. **a** Current density and relative luminance versus voltage. **b** Molecular structure of BCzVBi and the energy level diagram of the corresponding single layer device. **c** EL spectrum. **d** Photocurrent vs. current density in the low-voltage region and **e** in the high-voltage region. **f** Transient EL dynamics of the device after turn-off

outcompetes the first-order triplet decay processes and the fluorescence emission becomes directly proportional to the incident light flux exhibiting linear dependence. Precisely the same phenomenon is to be expected in electroluminescent devices when small-to-large concentrations of annihilating triplets are proportionally generated with increasing driving voltage (which increases current and therefore the triplet concentration), analogous to photochemical up-conversion, Kondakov and coworkers have previously shown that in OLEDs exhibiting triplet fusion, at low current density, because of low triplet exciton density, the luminance displays quadratic output behavior *vs* current density. In this regime the probability of triplets poised to annihilate is low, placing it in the weak annihilation limit. At higher current, the luminance exhibits linear dependence with increasing current density when the TF

process becomes dominant[9]. To confirm triplet fusion in the single-layer devices, photocurrent as a function of current density for the two different regions was measured. Figure 2d, e shows the low- and high-voltage regions for the BCzVBi single-layer device. Figure 2d shows the quadratic trend indicative of the TF mechanism leading to light emission, while Fig. 2e shows the linear dependence, indicative of direct formation of singlet excitons leading to emission. Moreover, fluorescent OLEDs with an active triplet fusion mechanism show a delayed EL after device turn-off due to the long lifetime of the triplet excitons[15,23–25]. Figure 2f shows the delayed EL decay of the BCzVBi single-layer device, further confirming that the EL originates from the triplets.

Next, we made a CzPA single-layer device. The device characteristics of the CzPA single-layer device are shown in Supplementary Fig. 2. The CzPA single-layer device displayed EL

at voltages as low as 2.35 V, with a bandgap energy of 3.17 eV. Similar to the BCzVBi single-layer device, the photocurrent versus the current density of the CzPA device also shows quadratic behavior for the low current region and linear behavior for the high current region, and the transient EL of the device displays microsecond delayed fluorescence, indicative of emission originating from triplet fusion.

These data clearly indicate that upon sub-bandgap electrical excitation, triplet excitons are present in these devices ultimately leading to delayed EL. As discussed previously, there are no other carrier transport layers present in these single-layer devices leading to the formation of CT excitons and that rules out the possibility of the presence of an exciplex state. Upon examination of these data, it appears that the voltage at the onset of EL for the single-layer devices corresponds to the sum of the Schottky barrier energy for hole injection from PEDOT and the first triplet excited state energy. Given the 5.2 eV work function of PEDOT, the Schottky barriers for hole injection for the CzPA and the BCzVBi single-layer devices are 0.51 and 0.18 eV, respectively. Adding the triplet energy of the two materials, that is 1.77 eV for CzPA and 1.94 eV for BCzVBi, to the Schottky barriers, we get energies of 2.28 and 2.12 eV, which are very close to the experimental values for the onset voltages of EL for the single-layer devices of 2.35 and 2.10 V, respectively.

To further investigate whether the sub-bandgap EL turn-on is observed in other material systems exhibiting triplet fusion leading to delayed fluorescence, more single-layer devices were examined using organic emitters exhibiting triplet fusion having a large singlet-triplet splitting: BDAVBi (4,4'-bis[4-(diphenylamino)styryl] biphenyl), DMPPP (1,1'-(2,5-dimethyl-1,4-phenylene)dipyrene) and TBADN (2-tert-butyl-9,10-di(naphth-2-yl)anthracene). All three devices showed EL at sub-bandgap voltages close to the sum of the values of their triplet energy and the hole injection Schottky barrier, similar to BCzVBi and CzPA. Moreover, in addition to a transient delayed electro-fluorescence, all these single layer devices show a quadratic dependence of the photocurrent on measured current density in the sub-bandgap region, indicating that the EL occurs via triplet fusion. The device structures and characteristics of the BDAVBi, DMPPP and TBADN single-layer devices are shown in Supplementary Figs. 3 and 4 and 5, respectively.

Given the strong evidence that all the organic molecules examined above feature EL and triplet fusion at sub-bandgap voltages, we decided to revisit rubrene. Rubrene is a well-known organic single molecule which shows P-type delayed fluorescence via triplet fusion with close to 100% efficiency[8]. The rubrene single-layer device was made with the following structure: ITO / PEDOT (30 nm) / Rubrene (40 nm) / Aluminum (100 nm), as shown in Fig. 3. Here, a bare aluminum electrode is used directly on top of the rubrene layer without an electron transport/ injection layer. To have a better alignment between aluminum work function (4.08 eV from vacuum) and the triplet level of rubrene (4.10 eV from vacuum), $Cs_2CO_3$ was not used for the rubrene single-layer device. Figure 3a shows a diode turn-on of about 1 V and an onset of EL at about 1.10 V. The results show that delayed fluorescence due to triplet fusion can be turned-on at voltages close to the triplet energy and triplet excitons can be formed without providing the bandgap energy for injection into the HOMO-LUMO levels. This finding suggests that full bandgap

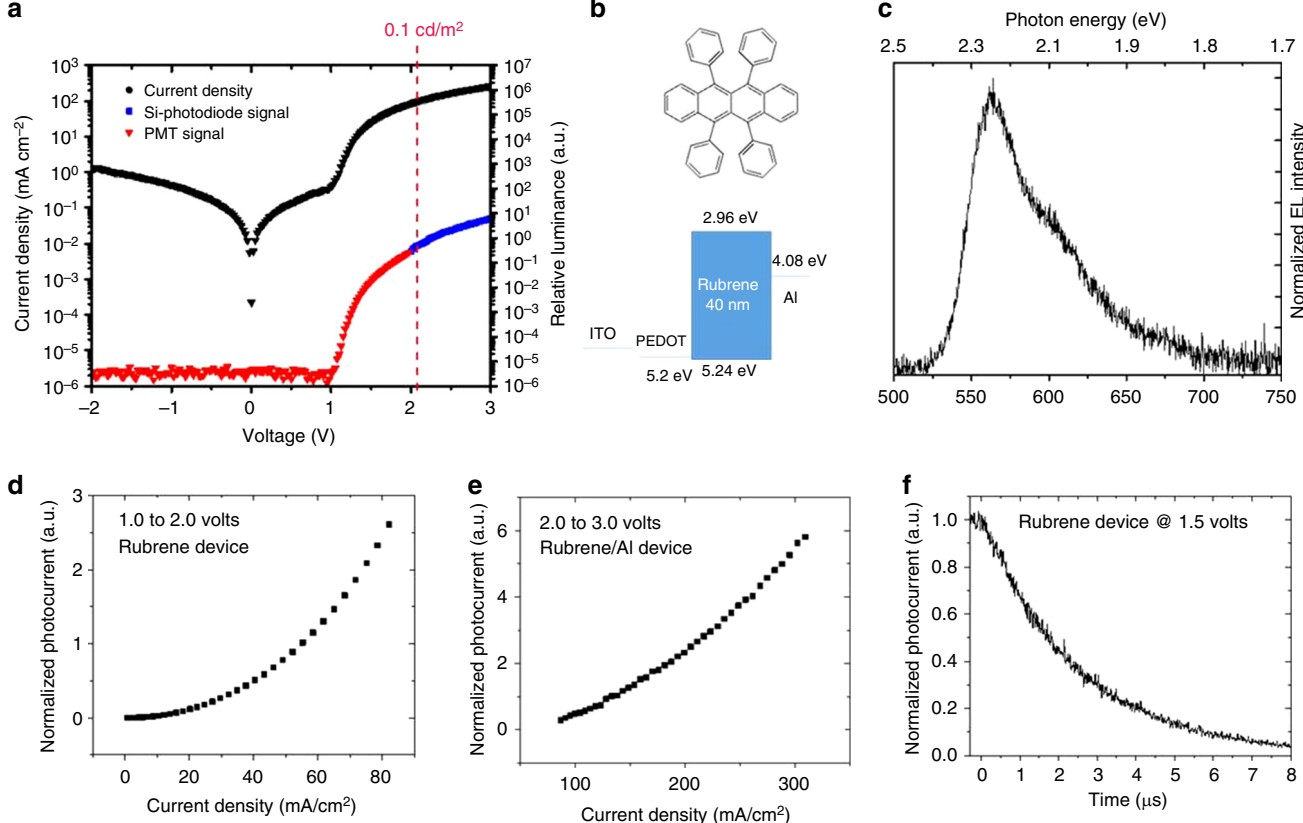

**Fig. 3** Single layer rubrene device. **a** Current density and relative luminance versus voltage. **b** Molecular structure of rubrene and the energy level diagram of the corresponding single layer device. **c** EL spectrum. **d** Photocurrent vs. current density in the low-voltage region and **e** in the high-voltage region. **f** Transient EL dynamics of the device after turn-off

voltages are not required for an OLED with emitters with sufficiently low triplet energies.

To further understand the exciton formation mechanism, rubrene single layer devices with cathode interlayers to facilitate electron injection were also fabricated for comparison. In a normal OLED, a cathode interlayer such as $Cs_2CO_3$ or LiF is usually used to lower the work function of the aluminum electrode and facilitate electron injection[26]. In contrast to conventional OLEDs where the turn-on voltage is lowered by introducing a cathode interlayer, a turn-on voltage increase was observed in the rubrene single layer OLED when a thin layer of $Cs_2CO_3$ or LiF is used as shown in Fig. 4. These results can be explained by the misalignment of the cathode work function with the triplet level of rubrene. When a cathode interlayer is inserted between the aluminum cathode and rubrene, the effective work function is lowered, resulting in a better energy alignment with the rubrene LUMO level rather than the triplet level. This misalignment with the triplet level and the better-alignment with the LUMO level results in a competition between electron injection into the LUMO level and the triplet level. As a result of introducing the cathode interlayer, the device operates closer a "normal OLED" where a full bandgap turn-on voltage is expected and that is the reason for the higher turn-on voltage.

Based on these results, we propose the following mechanism for triplet formation. When a sufficient bias voltage is applied to the device, holes can overcome the barrier and are injected from PEDOT to the HOMO level of the emitter molecules. Subsequently, in the presence of holes accumulating at the HTL/emitter interface, electrons from the cathode are injected to the triplet excited state of the emitter molecules generating triplet excitons, leading to sub-bandgap EL via triplet fusion. It should be noted that, under this conditions, electrons do not need to be injected to the LUMO frontier orbital to generate excitions. Therefore, for the single-layer devices, the onset voltage of EL would be at the voltages that can provide sufficient energy for holes to overcome the hole injection Schottky barrier ($\phi_{\text{h}}$) from PEDOT to the HOMO level of the TF molecule, and enough energy for electrons to be injected to triplets of the molecule, i.e., $V_{\text{on}}{\sim}\phi_{\text{h}+}T_1$, as shown in Fig. 5a. Supplementary Table 1 shows a list of molecules studied in this work with the corresponding single-layer devices exhibiting sub-bandgap EL due to TF having the structure: ITO / PEDOT (30 nm) / emitters (40 nm) / $Cs_2CO_3$ (1 nm) / Al (100 nm).

**Multiple layer OLEDs**. While very low turn-on voltages are demonstrated in the single layer devices, the EL efficiencies are inherently very low because the recombination zone is close to the metal electrode interface resulting in luminescence quenching. Building upon the above results, a high efficiency sub-bandgap OLED making use of TF with triplets directly formed by charge injection was fabricated. To design such a device, the hole injection barrier must be small and electrons are to be readily injected to the triplet level of the emitter from the ETL. Based on these criteria, we fabricated the following device using CzPA as the host and BCzVBi as the emitter: ITO (50 nm) / TAPC:MoO$_3$ 1:0.1 (30 nm) / TAPC (20 nm) / CzPA (3 nm) / CzPA:BCzVBi 1:0.1 (30 nm) / BCzVBi (3 nm) / BPhen (30 nm) / $Cs_2CO_3$ (1 nm) / Aluminum (100 nm). Here TAPC is 1,1-bis[[di-4-tolylamino)phenyl]cyclohexane and is used as the hole transport layer and BPhen is bathophenanthroline and is used as the electron transport layer. Thin films of CzPA and BCzVBi located at transporting layer sides function as hole or electron injection facilitating layers. The device characteristics are shown in Fig. 6. While BCzVBi and CzPA have

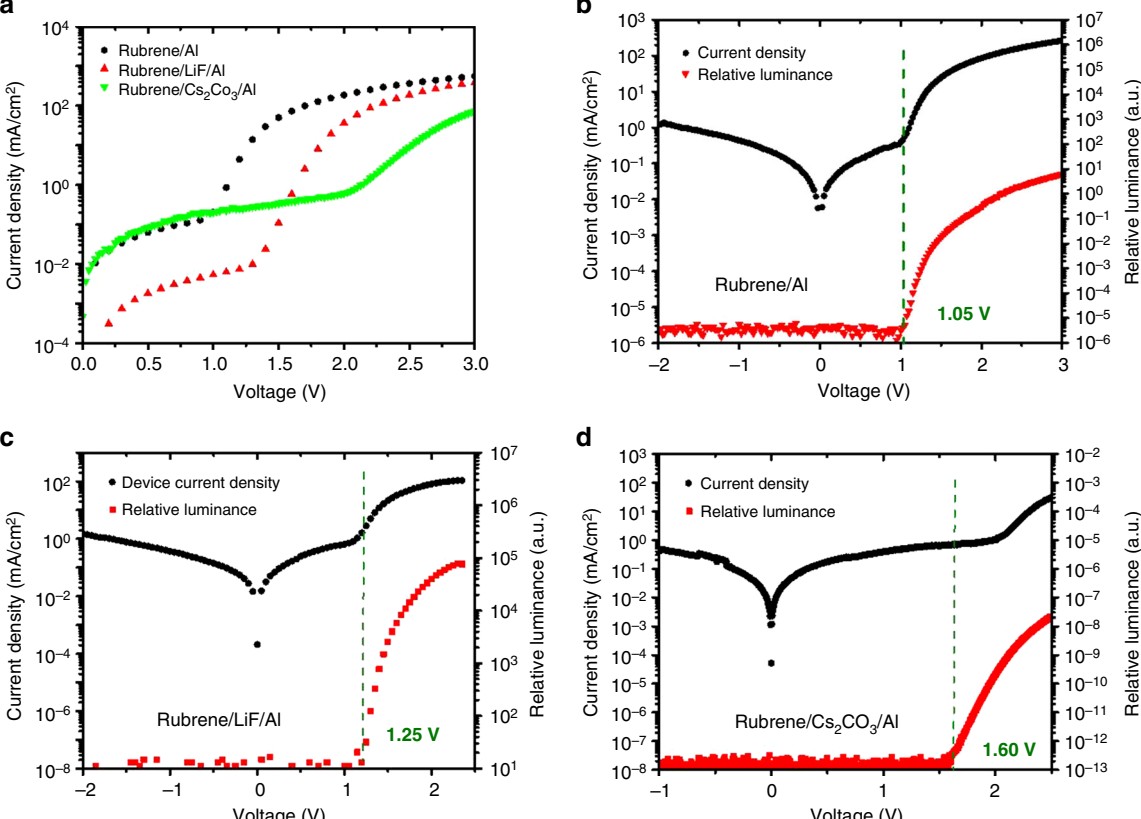

**Fig. 4** Electrical characteristics of different rubrene devices. **a** Current density as a function of voltage of rubrene single-layer OLEDs. **b–d** Current density and relative luminance versus voltage of the devices with Al, LiF/Al and $Cs_2CO_3$/Al structures

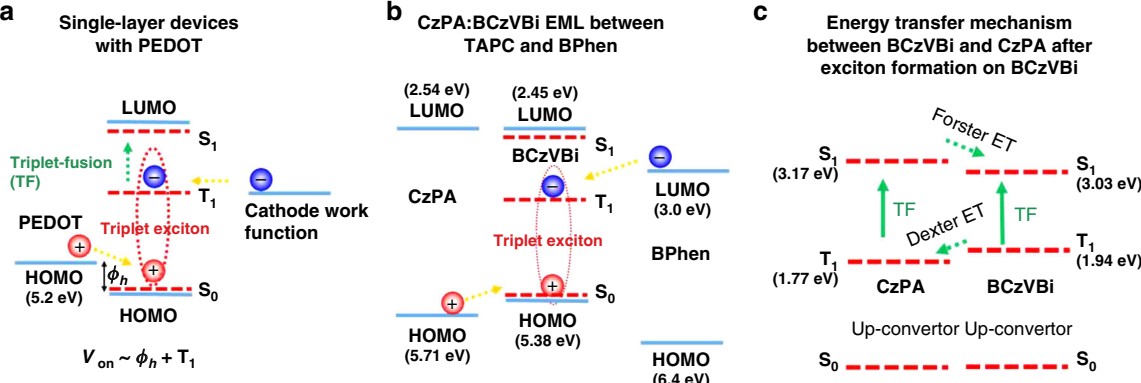

**Fig. 5** Direct triplet exciton formation mechanisms. **a** Direct triplet exciton formation in single-layer devices with the structure ITO/PEDOT/Emitter/Cathode. **b** Direct triplet exciton formation on BCzVBi molecule, in CzPA:BCzVBi EML placed between TAPC and BPhen as HTL and ETL, respectively. **c** Energy transfer mechanism and interaction between CzPA and BCzVBi

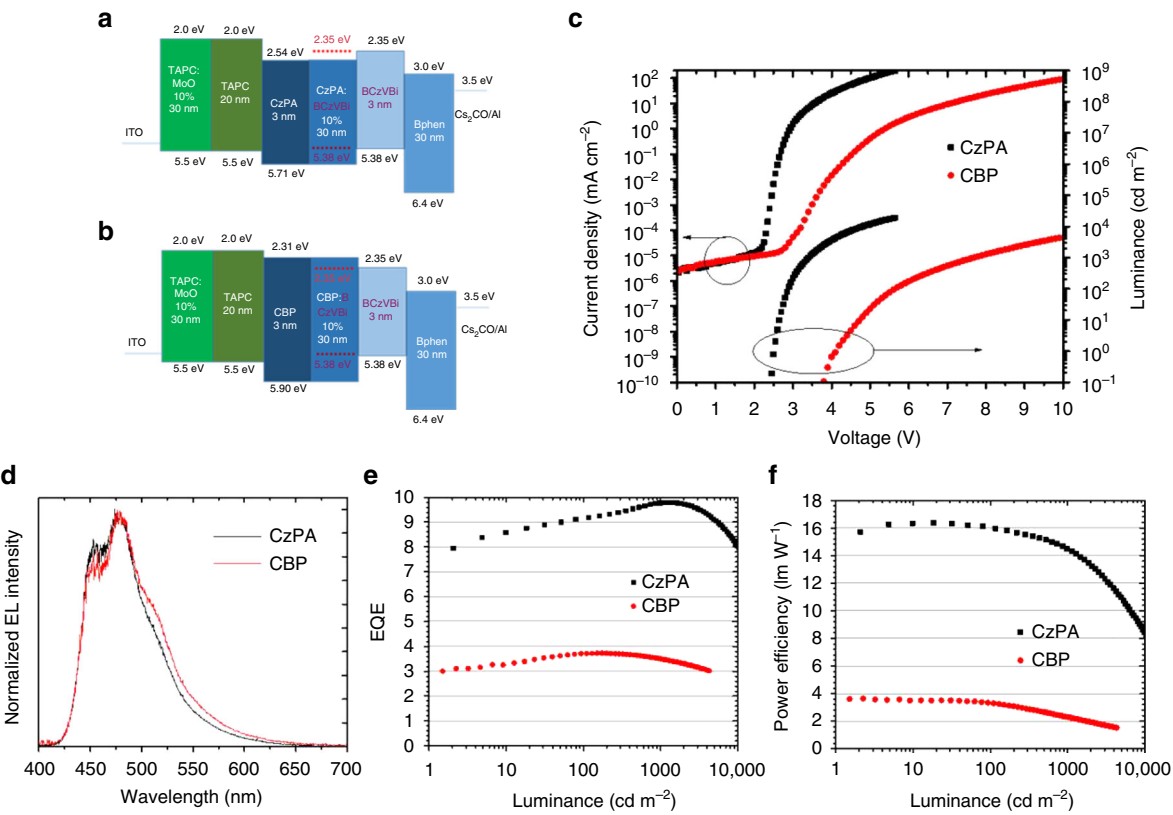

**Fig. 6** Multilayer OLEDs. Performances of the devices using CzPA and CBP as hosts vs the optimized device using CzPA as host. **a, b** Device structures. **c** Current density and luminance versus voltage of the devices. **d–f** EL spectra, EQE-luminance characteristics and power efficiency-luminance characteristics of the devices

HOMO-LUMO bandgaps of 3.03 and 3.17 eV, respectively and the device shows EL emission with photon energies up to 3.0 eV. The onset of EL takes place at 2.1 V (Supplementary Fig. 6a), which is similar to the BCzVBi single-layer device. The high efficiency device achieves a luminance of 0.1 cd/m² at 2.4 V, 100 cd/m² at 2.9 V and 1000 cd/m² at 3.4 V, respectively, as shown in Fig. 6a. The EL spectra of the optimized device indicated no change as a function of voltage, as shown in Figure S6b.

The structure of the multi-layer device is the result of many optimization runs to achieve the best charge balance for maximum efficiency. The results of the optimization runs are

shown in Supplementary Figs. 7, 8 and 9. All OLEDs in the optimization runs show a luminance of 0.1 cd/m² at 2.4 V with their corresponding EL spectra displaying the emission from BCzVBi only. Further details on optimization runs are included in the supporting information section.

The onset of EL at the same voltage of 2.1 V for both the single-layer BCzVBi device and the multilayer device suggests that the initial triplet exciton formation occurs on the BCzVBi molecules and that BCzVBi is responsible for EL at initial sub-bandgap voltages, as shown in Fig. 5b. Upon formation of triplet excitons on the BCzVBi molecules, three processes can occur which would

ultimately lead to the generation of singlet excitons on BCzVBi at sub-bandgap voltages, as shown in Fig. 5c. First, BCzVBi triplet excitons can annihilate with each other and create radiative singlet excitons on BCzVBi molecules, like the case in the BCzVBi single-layer device. Second, BCzVBi triplet excitons can transfer their energies to CzPA triplet excitons via Dexter energy transfer and the CzPA triplet excitons can then annihilate and create a singlet exciton on CzPA molecules. The singlet excitons on the CzPA molecules would then transfer their energy by Förster energy transfer to the singlet excitons on BCzVBi, due to the spectral overlap between CzPA emission and BCzVBi absorption. Third, CzPA and BCzVBi triplet excitons can hetero-annihilate and ultimately lead to formation of singlet excitons on BCzVBi. The formation of triplet excitons with direct injection of the electrons into the LUMO level of the TF molecules leads to sub-bandgap EL from our state of the art blue fluorescent device and its high luminance at such low voltages.

To investigate the effects of triplet energy on the device operation, a high $T_1$ energy level host 4,4′-Bis(carbazol-9-yl) biphenyl (CBP) was used to replace CzPA in the same device structure. CBP is a host material with a high triplet energy of 2.6 eV, that is known not to exhibit any delayed fluorescence nor triplet-fusion[25,27]. Two important points should be noted here. First, if the operating voltage of the OLED is determined by the triplet energy, it is expected the operating voltage of the CBP device is higher than that of the CzPA device. Second, since CBP is not a TTA host, the quantum efficiency is expected to be lower than the corresponding CzPA device. Figure 6 shows the device structure and performance of the CBP device versus the optimized CzPA device. We emphasize that the only difference between the two devices is the host material and everything else is the same. Again, two things should be noted here. First, the device using CBP as the host has a much larger operating voltage compared to the optimized CzPA device. For example, the operating voltage at 1,000 cd/cm² is 3.5 V for the CZPA device compared to 7.6 V for the CBP device. In addition, we notice that the onset of EL is 3.7 V for the CBP device compared with 2.4 V for the CzPA device. The results demonstrate that the triplet energy of the host has a significant effect on the device operating voltage as we predicted. Second, the EQE of the CzPA device is almost 10% while the EQE of the CBP device is less than 4%. Again, these results confirm the non-TF nature of the CBP fluorescent device. The combination of the lower operating voltage and the higher EQE value resulted in the CzPA device having a power efficiency more than 4 times higher than that of the normal flourescent device.

To further demonstrate the significance of our present work, we compare the performance of our TF OLEDs with phosphorescent and thermally activated delayed fluorescent (TADF) blue OLEDs. Table 1 summarizes the device performance comparison of our work and some of the best phosphorescent and TADF OLEDs with similar CIE color coordinates. Although TADF and phosphorescent OLEDs can achieve very high values of EQE and PE at very low luminance, at higher practical luminance ranges (>500 cd/m²) they show a significant efficiency roll-off. Our optimized fluorescent device does not show any efficiency roll-off at a luminance up to 2,000 cd/m², and owing to its low operating voltages, it shows a higher power efficiency compared to all other TADF and phosphorescent devices with similar spectral color coordinates. Our device achieves a maximum EQE of 9.8% and PE of 16.4 lm/W and an EQE of 9.8% at 1000 cd/m² and PE of 14.5 lm/W at 1000 cd/m². These results suggest that at higher current densities the triplet fusion process becomes increasingly efficient which is akin to photochemical up-conversion processes where combinations of high sensitizer and annihilator concentrations along with strong light excitation leads to the highest possible efficiencies that can be realized in a triplet fusion composition[28].

## Discussion

This work demonstrated that triplet excitons can form under low driving voltage around the triplet exciton energy of the materials and subsequently generate singlet excitons through TF process and emit light. The onset voltage can be as low as half of the molecular HOMO-LUMO bandgap. The direct triplet formation occurs in wide-bandgap blue-emitting single organic molecules as well as narrower band-gap yellow emitting organic molecules. Based on these findings, a state of the art blue fluorescent OLED with a power efficiency higher than some of the best phosphorescent and TADF blue OLEDs at high luminance was achieved. This discovery indicates that while the internal quantum efficiency is half of that of phosphorescent or TADF OLEDs, the efficacy at high luminance can actually be higher due to the lower applied voltages. There are two advantages of these devices. First, the operating voltage of these fluorescent OLEDs is lower than conventional devices, which leads to much lower power consumption. If the triplet energy of the emitter is half of the singlet energy, the operating voltage of the resulting device might be only half of the conventaional OLED. Even though the quantum efficiency of the triplet-fusion device is half of that of the phosphorescent device, the resulting power efficiency in lumens per watt will be the same. Second, the reduced operational voltage can lead to a longer operational lifetime of the device. Further experiments will be carried out to validate the point.

## Methods

**Materials**. TAPC, BPhen, BCzVBi, DMPPP, BDAVBi and rubrene were purchased from Lumtec Technology Corp. CzPA was purchased from ChangChun Tuo Cai Technology Co., Ltd. ITO thickness for all the substrates used was 50 nm thick.

**Table 1 Performance Comparison**

| Emitter | V @ 1 cd/m² (V) | V @ 500 cd/m² (V) | V @ 1000 cd/m² (V) | Max PE (lm/W) | PE @ 500 cd/m² (lm/W) | PE @ 1000 cd/m² (lm/W) | Max EQE (%) | EQE @ 500 cd/m² (%) | EQE @ 1000 cd/m² (%) | CIE (x,y) | Ref. |
|---|---|---|---|---|---|---|---|---|---|---|---|
| BCzVBi | 2.50 | 3.2 | 3.4 | 16.4 | **15.1** | **14.5** | 9.8 | 9.6 | 9.8 | (0.15,0.21) | This Work |
| DTPDDA | 3.0 | 5.0 | 5.8 | 30.4 | 10 | 9.8 | 22.3 | 11 | 10.6 | (0.15,0.20) | 30 |
| 34TCzTTrz | 3.8 | 7.0 | 9.8 | 15.0 | N/R | N/R | 10.3 | <5 | <1 | (0.16,0.20) | 31 |
| CzAcSF: TBPe | 3.0 | 5.0 | 5.0 | 23.4 | N/R | 9.7 | 15.4 | N/R | 10.7 | (0.15,0.23) | 32 |
| DDCzTrz | 4.0 | 5.9 | 8.0 | 23.6 | 4.0 | N/R | 18.4 | 5.0 | N/R | (0.16,0.21) | 33 |
| FCNIrpic | 4.0 | 6.0 | 8.0 | 29.8 | 11.9 | 10.2 | 18.9 | 17.7 | 16.7 | (0.14,0.21) | 34 |
| Ir(dbfmi) | 2.56 | N/R | 4.74 | 35.9 | 10 | 6.3 | 18.6 | N/R | 6.2 | (0.15,0.19) | 35 |

Summary of our device and some of the best TADF and phosphorescent OLEDs with similar color coordinates reported in literature
N/R not-reported

**Optical properties.** For ellipsometry measurements a Woollam M2000 ellipsometer was used, and the data was fitted using WVASE software.

**Singlet and triplet energies.** Molecular triplet energies were experimentally estimated via phosphorescence spectra recorded at 77 K, Supplementary Fig. 10. The material of interest was dissolved in a glass-forming, heavy-atom containing solvent of 40 wt.% toluene, 40 wt% dichloromethane, and 20 wt.% iodobenzene. The solution was cooled to 77 K in a transparent liquid-nitrogen Dewar. Time-gated phosphorescence spectra of the molecules were collected with an LP920 laser flash photolysis system (Edinburgh Instruments) using a wavelength-tunable Vibrant 355 LD-UVM Nd:YAG/OPO system (OPOTEK) as the excitation source (1.0–2.6 mJ/pulse). Emission spectra were collected with an iStar ICCD camera (Andor Technology), which was controlled by the LP900 software program (Edinburgh Instruments). The gate delay after excitation into the wavelength of each molecule's respective absorbance maximum was 200 μs, and the gate width for data collection was 200 μs. Time-gated phosphorescence spectra reported herein are the average of at least 512 laser flashes.

Molecular singlet energies were estimated from solution-phase absorption and emission spectra, Supplementary Fig. 11. Absorption spectra were acquired using an Agilent 8453 diode array spectrophotometer. Room temperature and 77 K static emission spectra were collected with a FLS980 fluorometer (Edinburgh Instruments) equipped with a 450 W Xe arc lamp and a Peltier cooled PMT (R928P Hamamatsu). The 77 K emission spectra were collected using a 2-MeTHF optical glass cooled in the same manner detailed above. All static emission spectra were corrected for detector response.

**Energy levels.** Molecular HOMO energy levels of interest were estimated from redox potentials measured by cyclic voltammetry (CV) using a CH Instruments CHI650E Electrochemical Analyzer/Workstation. Measurements were carried out in a single electrochemical cell with a typical three-electrode configuration contained within an inert atmosphere glovebox (MBraun). The cell configuration consisted of a platinum disk working electrode, platinum wire counter electrode, and a Ag/AgNO₃ reference electrode. Analyte solutions were prepared in degassed dry dichloromethane containing 0.1 M tetrabutylammonium hexafluorophosphate (TBAPF₆, recrystallized from ethanol) as the supporting electrolyte and analyzed at a scan rate of 0.100 V/s. As an internal standard, ferrocene exhibited a reproducible oxidation potential of 0.01 V in the same electrochemical cell and solvent. The electrochemical potentials determined from cyclic voltammetry were converted to HOMO values using a modified ionization potential for ferrocene in dichloromethane of 4.89 eV below vacuum[29].

**Device fabrication and characterization.** Fabrication and evaporation of all layers was carried out under vacuum pressure of below $2e^{-6}$ torr. The evaporation rate was between 1 and 2 A/s. Electroluminescent spectrum was collected using an OceanOptics spectrometer and calibrated using a standard lamp. A PMT (Hamamatsu R6358) was used to detect low-intensity EL and transient EL. A Si-photodiode, OSI Optoelectronics PIN UV-100, was used to detect high-intensity EL. Luminance was measured by calibrating photocurrent using a lumin-gun. A semi-hemispherical lens was placed on the substrate to extract substrate-trapped light for all luminance-voltage-photocurrent measurements.

**Transient EL measurement.** An Agilent function generator, 3320 A, was used for transient EL (TrEL) measurements. A 100 micro-second long pulse following a reverse bias of 5 V was applied to devices for the TrEL measurements. Two 50.00 ohms resistors were used with the PMT and the OLED devices, as shunt resistors, to minimize RC response in TrEL measurements.

## Data Availability

All data generated or analysed during this study are included in this published article (and its supplementary information fles). Supplementary information is available in the online version of the paper. Correspondence and requests for materials should be addressed to F.S.

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

## Acknowledgements
The authors acknowledge the support of the North Carolina State University Game-Changing Research Incentive Program (GRIP).

## Author contributions
A.S. and F.S. conceived and designed the experiments. A.S., C.D. and D.S. fabricated and characterized the OLEDs. C.M.P. made CV measurements and the PL spectral measurements. A.S., C.D., D.S., L.Z., C.M.P., A.B., F.N.C. and F.S. analyzed the data and co-wrote the paper. F.N.C. and F.S. supervised the work.

## Additional information

**Competing interests:** The authors declare no competing interests.

