## [Peer Review File · Nature Communications]

Reviewers' comments:

Reviewer #1 (Remarks to the Author):

This paper describes a new effect in OLED devices, namely direct injection into the triplet level of the organic emitter material. Previously, as the authors state, the assumption was that a voltage comparable to the HOMO-LUMO gap (basically the singlet energy) was required to inject charges. This paper shows that injection into the triplet states can be accomplished and optimized by judicious selection of the device layers. This capability is especially important for blue-emitting OLEDs which rely on triplet-triplet annihilation or fusion to generate the high energy singlet excitons. This work is clearly written and represents a significant advance. It should be published after the comments listed below are addressed.

1) On p. 3, the authors state that sensitized triplet fusion EQEs are "typically less than 1%". This is not quite accurate: ref. 16 shows an EQE of 4%, and subsequent work by Lee and coworkers (Adv. Mater. 30, 1804850 (2018)) has shown some success by spatially segregating the triplet formation and upconversion. Of course, the results of this paper suggest that this segregation may not be necessary, but they should recent acknowledge progress in this area.

2) I was surprised that the CBP control devices worked so well (Fig. 4e), but these curves are for different driving voltages, correct? The voltages used for the EQE curves should be stated in the caption, otherwise the uncaredful reader might assume that triplet fusion must be going on in the CBP devices as well.

3) The whole reason that triplet fusion is used for blue OLEDs is because it enhances the device stability by avoiding the formation of high energy states via singlet-singlet or singlet-polaron annihilation, which lead to chemical changes. I could find no information on the stability of the devices, even in relative terms. The authors should add a brief discussion of stability (e.g. hours to 80% output, etc.), at least so the field knows whether they must be optimized.

4) Overall, the ability to drive EL by triplet injection is surprising but also general. Why was this not observed before? Is it just that previous workers did not look hard enough at low voltages? The authors say their efficiency was "very low" in unoptimized devices, and then show that it is quite large when they optimize using a multi-layer structure. How much of an increase does the optimization provide? Factor of 10, 100, ?

5) The "mechanism for triplet formation" on p. 8 looks the same as the standard mechanism for singlet exciton formation. Is there anything special about triplet formation via injection, other than matching HOMO/LUMO levels? What about relative efficiencies for triplet versus singlet formation, given the voltage requirements are met? Is there any kinetic issue with triplets that is absent for singlets?

6) p. 11 "To further investigate the significance of our present work" I think they mean "To further demonstrate the significance..."

Reviewer #2 (Remarks to the Author):

This paper describes the use of triplet fusion to enhance the efficiency of OLEDs beyond the 25% limit imposed by a fluorescence. If only the singlet fraction of excitons generated in the electroluminescent process is used the limit is 25%, but if the triplets are utilized as well the efficiency can be driven to 100% (here this is the internal efficiency). Effective utilization of triplets was first demonstrated with phosphorescent dopants and later with TADF emitters. The approach explored here is not a new one, but their efficiencies are higher than those reported previously. A

group at Kodak reported the use of triplet fusion to enhance OLED efficiencies in 2009. The approach described in the 2009 work is the same one applied here, but the present paper uses better materials and device structures to improve their efficiencies to over 10% (EQE). While this is an advance, it falls well short of the 20% EQEs that both heavy metal based and TADF phosphors have achieved. The paper is not appropriate for Nature Comm on the basis of their device efficiency or basic mechanism for collecting triplets. The authors make the point that their devices have "a power efficiency higher than some of the best phosphorescent blue OLED". These authors know full well that such a claim is meaningless without comparing spectra. Their EL spectrum has a marked green tail. This impacts the luminance markedly and to compare their devices for their power efficiencies (lm/W) with no mention of spectral purity is misleading. I have not pulled all of the references, but at least some of the devices they compare to are display level blue compounds, making their inherent luminance low do to a poorer match to human photopic response. Moreover, they are comparing their devices, which have low driving voltage, to devices that were not designed to have low volatage. If they had compared their OLEDs to blue phosphorescent OLEDs with conductivity doped HT and ET layers they would not have seen such a difference. They would have found that the heavy metal phosphor based device outperforms theirs, even before they match spectra.

There is another issue that the authors seem to have missed or at least chose not to fully explain to the reader. In their device they go to lengths to ensure that the hole/electron recombination generates the triplet directly, without going through the singlet or forming the singlet and triplet in a 1:3 ratio. The triplets then annihilate to form a singlet, which gives rise to emission. The math here is pretty simple. If they are totally useful they will be limited to a theoretical internal efficiency of 50%, and the 10% EQE they see here is a dead-end. They cannot improve beyond this level and report here the best device that will ever be reported. The unfortunate thing is that this EQE is well below that of the commercial devices or even average devices. They have chosen to push the power efficiency angle to make their devices more competitive, but as I said above, there are other ways to lower voltage and thus improve power efficiency. They also make a claim that their devices have low roll-off as they are made brighter. Again, this is a bit of smoke and mirrors. The plot EQE vs. luminance and say that their device is wonderful compared to a CBP based fluorescent device. On a fractional drop bases the two devices are the same, even though one is singlet only and the other is populating exclusively form triplet fusion. Moreover, their plot makes it tough to compare to phosphorscent OLEDs. The normal plot is EQE versus current density. If they replotted their data you would see that the roll-off is pretty typical and that of you look in the literature you will find devices that have much lower roll-off than theirs.

Based on the previous two paragraphs you would come to the conclusion that I don't think this paper is Nature Communications material. There is a part of this paper that rises to your standard though and may be sufficient to justify publishing it in Nature Communications. This has to do with the direct formation of triplets. I heard Steve Forrest talk about resonant carrier recombination into the triplet exciton some time ago, but I have dug through the literature and have not been able to find any literature report of this. I found a patent, but it has no experimental data, and is not a peer reviewed report in any even. This paper represents the first clear example of being able to directly form the triplet exciton in electroluminescence with high efficiency. The physics of this is very interesting and is discussed in this paper. In my opinion the paper should be built around this process. It will involve a total rework of the paper to shift the emphases away from competing with phosphorescent OLEDs to one of developing new routes to higher efficiency in a device that has the potential to achieve 100% internal efficiency and not one that will dead-end at 50%. If they want to rework their paper along this line I would be happy to rereview it, but in this form, it is not suitable for publication.

Issues for the authors to consider in a revised manuscript:

- This is a scientific paper, not a trade journal. The reader should not need to go to the SI to find the structures of the compounds being used here. The first time a compound is referenced in the

text the structure should be given. A simple scheme with all of the materials in the paper needs to be added.

- In a revised manuscript I would not sue such heavy reliance on the SI. Half of the paper (pages 4-8) rely totally on the SI. This is crazy. The authors should move some of the figures in the SI into the paper. It is not fair to expect the reader to flip back and forth between the paper and the SI so much. The SI is supporting, after all, and if the SI is required to really understand the paper it is no longer "supporting"
- In the experimental the authors give complete detail on how they do their electrochemistry, but do not talk about how they convert their potentials (relative to ferrocene) to HOMO energies. There are a number of good papers on this, and they should reference one of them or explain what they did to convert the numbers and why. Their HOMO values are more or less what I expected so I think that this is a matter of the experimental discussion not being complete. There is a much bigger problem with their LUMO values. There is no mention of where these values came from and nearly all of them are wrong, with some off by 1 eV or more. Looking at their values I am guessing that these are optical LUMOs, derived by adding the optical gap to the HOMO energy. The problem is that this is not the LUMO energy. The LUMO energy is estimated electrochemically or determined by IPES. To give you an example, they quote LUMO values for TAPC, CBP and BPhen of -2.0, -2.9 and -3.0, but the correct values are -0.05, -1.8 and -2.0. They miss on both the absolute values and the relative ones as well. They need to explain the origin of their numbers and I suggest they do some measurements or literature digging to get the right values.
- Taking the discussion of HOMO and LUMO values above into account, there is a scientific problem they should address. In their single layer devices it is not a problem for eh electrode to deliver electrons at a potential that matches the triplet, but the organic materials are not metals. Their HOMO and LUMO values fixed. That means that in the multilayer device, you can still form the triplet directly, but the voltage benefit should be gone. They clearly show this for a fluorescent device, but why is the same not true for a triplet fusion based device? With their (wrong) LUMO values it looked like you could still inject into the triplet at low voltage, but this should not be the case.
- The authors use the ordinary and extraordinary extinction coefficients to determine the alignment of BCzVBi in a neat thin film. There are two problems with this. The analysis is only useful for single layer devices, when the material is used as a neat thin film. The devices where the alignment issue is important are the ones where it is present as a dopant. The second problem is that this is not the best method for determining the dopant alignment. Angle dependent PL measurements are far more accurate and have the benefit that they can be used to probe the alignment of a dopant in a film directly. Also, what is the alignment of CzPA? It seems incomplete to only consider a single material in their alignment studies.
- The authors continuously refer to the brightness of their OLEDs by the "photocurrent". I am assuming that they are talking about the photocurrent produced by their photodetector. Since they always give these values as arbitrary units, they should just call it brightness or intensity, not photocurrent. Even better would be to convert the axis to luminance. They do this on some of their plots, why not all of them. Instead of photocurrent change the axes to cd/m².
- Why don't the CzPA EL spectra in Figures S2 and S3 match?
- Nature Communications is a general science journal not an optics one. It would be useful to add a sentence or two at line 117 to explain to the reader why the "photocurrent" is quadratic with triplet density at low brightness but changes over to linear at high brightness. Why isn't it quadratic all the way up? This is a topic that Castellano has discussed at some length in his upconversion work and a couple of sentences here to better explain the physics would be useful for a reader that is new to this area.

Response to Review Comments

Ms. Ref. No.: NCOMMS-19-04214

We are grateful to the reviewers for their comprehensive review and comments. We have revised our present manuscript in the light of their valuable suggestions and comments, and believe that the quality of our revised manuscript has significantly been improved and the manuscript should be ready for publication.

We believe that it is important to reiterate the importance of this work. This work is important in terms of the fundamental physics and chemistry of the devices. We discovered that efficient light emission in triplet fusion OLEDs is observed at voltages below the bandgap. Based on our device data, it appears that the formation of triplets is a direct result of carrier injection without injection of electrons into the frontier LUMO orbital. As a result, if the triplet energy is a half of the singlet energy, the drive voltage of a triplet fusion OLED can be a half of the voltage to drive a phosphorescent OLED (PHOLED) or TADF OLED. This is totally unexpected. Triplet states do not exist in ground state and they exist only when the molecules are in their excited states. It is therefore not expected that triplets can be formed by direct charge injection. If our conjecture is correct, it is a ground-breaking science in OLED device physics.

In addition to the new generation mechanism of triplet states by charge injection, our discovery has also an important technological implication. Since the discovery of PHOLEDs, OLED has advanced to a point that it has become a dominant display technology in mobile devices and TVs. However, while both red and green PHOLEDs are widely used, blue PHOLEDs still have a serious lifetime issue after almost 20 years of research. The major problem of blue PHOLEDs is the high triplet energy and its long exciton lifetime, leading to triplet-triplet annihilation generating much higher energy excited states and breaking chemical bonds. Even with the recent discovery of TADF emitters, the blue OLED lifetime still remains a great challenge. Therefore, blue fluorescent OLEDs are used in commercial display even though they have a lower efficiency. Our low voltage triplet-fusion OLEDs might be a technological breakthrough in OLED technology. If the drive voltage is half of the standard OLEDs, the power efficiency might be as high as conventional PHOLEDs or TADF OLEDs. The advantage of the triplet-fusion OLEDs is that its triplet energy is about half of the triplet energy of the emitters in PHOLEDs. Because of the lower excited state energy, it is expected the lifetime will be significantly better.

Therefore, we believe this is the breakthrough finding of the article which is worthy of publication in Nature Communications.

Reviewer #1 Comments

This paper describes a new effect in OLED devices, namely direct injection into the triplet level of the organic emitter material. Previously, as the authors' state, the assumption was that a voltage comparable to the HOMO-LUMO gap (basically the singlet energy) was required to inject charges. This paper shows that injection into the triplet states can be accomplished and optimized by judicious selection of the device layers. This capability is especially important for blue-emitting OLEDs which rely on triplet-triplet annihilation or fusion to generate the high energy singlet excitons. This work is clearly written and represents a significant advance. It should be published after the comments listed below are addressed.

Comment 1 On p. 3, the authors state that sensitized triplet fusion EQEs are “typically less than 1%”. This is not quite accurate: ref. 16 shows an EQE of 4%, and subsequent work by Lee and coworkers (*Adv. Mater.* 30, 1804850 (2018)) has shown some success by spatially segregating the triplet formation and upconversion. Of course, the results of this paper suggest that this segregation may not be necessary, but they should recent acknowledge progress in this area.

Response: Thank you for pointing out the important update. We have changed the statement from “the electroluminescence efficiency is less than 1%” to “the electroluminescence quantum efficiency is typically low” on page 3. We also added the subsequent work by Lee and coworkers (*Adv. Mater.* 30, 1804850 (2018)) into the reference. The sensitized triplet fusion EQEs are still low because there is a quenching path from the singlet state to the CT states. Therefore, the intrinsic EQE of this process is low.

Comment 2 I was surprised that the CBP control devices worked so well (Fig. 4e), but these curves are for different driving voltages, correct? The voltages used for the EQE curves should be stated in the caption, otherwise the uncaredful reader might assume that triplet fusion must be going on in the CBP devices as well.

Response: The data we obtained are from the current-voltage-luminance

measurements. In most cases, the EQE data are presented in a format of the EQE-Luminance (EQE vs L) or EQE-Current density (EQE vs J). Fig. 4e shows the EQE-Luminance plots, not the EQE-Voltage curves. The voltage value can be determined by tracing the corresponding values of the luminance or current density values from Fig. 6e to Fig. 6c.

Comment 3 The whole reason that triplet fusion is used for blue OLEDs is because it enhances the device stability by avoiding the formation of high energy states via singlet-singlet or singlet-polaron annihilation, which lead to chemical changes. I could find no information on the stability of the devices, even in relative terms. The authors should add a brief discussion of stability (e.g. hours to 80% output, etc.), at least so the field knows whether they must be optimized.

Response: Thank you for the comment. Because of the lower excitation energy, it is expected that triplet fusion in blue fluorescent OLEDs has a better operating stability compared with the PHOLEDs. However, our work here is not aiming at investigating of the effect of triplet fusion on the device operating stability. The focus of the work is on the operation of triplet fusion OLEDs and the triplet generation mechanism. The lead author worked in the OLED industry for more than 14 years and determine the intrinsic OLED operating lifetime is tricky. To obtain good device lifetime, material purity is critically important. In this work, both the host and emitter materials used are purchased from commercial sources and they are very expensive. Purifying these materials by vacuum sublimation would be impractical because of the very low yield. As such, we cannot use ultra-purity materials for the present study and the device lifetime might not be meaningful. Nevertheless, we did the device lifetime measurements on the devices presented in this work and the data for both CzPA and CBP host devices are shown in the following figure. The starting luminance is 1000 cd/m^2 , and the LT50 of CzPA and CBP host devices are 4 hours and 1.5 hours, respectively. It should be noted that the device structure used was designed for high efficiency devices and not for stability studies, and therefore the device lifetimes are not good. Specifically, the hole transporting material TAPC used in this work is

known to give poor operating stability due to the formation of trap at the HTL/emitter interface. Nevertheless, the better lifetime of the CzPA devices is demonstrated. To carry out the actual lifetime study, a different device structure should be used and it is beyond the scope of this work.

Comment 4 Overall, the ability to drive EL by triplet injection is surprising but also general. Why was this not observed before? Is it just that previous workers did not look hard enough at low voltages? The authors say their efficiency was “very low” in unoptimized devices, and then show that it is quite large when they optimize using a multi-layer structure. How much of an increase does the optimization provide? Factor of 10, 100, ?

Response: Thank you for the valuable comment. In this work, we made both the single-layer device and the optimized multi-layer device for different purposes. The single layer devices were used to evaluate the turn-on characteristics for each pure material. Without the carrier transport layers to confine the excitons within the emissive layer, the luminescence is heavily quenched at the interfaces. For example, the EQE-Luminance figure of the BCzVBi single layer device below shows the EQE is around 1%, On the other hand, the optimized multilayer devices with balanced hole and electron transport shown in the manuscript show an EQE of about 10%.

Comment 5 The “mechanism for triplet formation” on p. 8 looks the same as the standard mechanism for singlet exciton formation. Is there anything special about triplet formation via injection, other than matching HOMO/LUMO levels? What about relative efficiencies for triplet versus singlet formation, given the voltage requirements are met? Is there any kinetic issue with triplets that is absent for singlets?

Response: For a normal fluorescent OLED, holes and electrons are injected into the HOMO/LUMO energy levels of the emissive molecule leading to the formation of singlets and triplets in a statistical ratio of 1:3. As a result, it is expected a voltage corresponding to the full bandgap (full LUMO-HOMO bandgap) is required. Based on the results of our triplet fusion OLEDs, we proposed triplets are formed directly by charge injection without going through the LUMO energy level and electrons do not need to overcome the barrier of LUMO energy level from the ETL to the EML. This is totally not expected in an OLED as the triplet level is not an electronic energy state into which electrons can be injected. We conjecture that due to the accumulation of holes at the interface, electrons with an energy corresponding to the triplet energy level can be injected to the emitter to form triplets.

Comment 6 p. 11 “To further investigate the significance of our present work” I think

they mean “To further demonstrate the significance...”

Response: Thank you for the comment. The statement has been changed to “To further demonstrate the significance of our present work” in the manuscript.

Reviewer #2 (Remarks to the Author):

This paper describes the use of triplet fusion to enhance the efficiency of OLEDs beyond the 25% limit imposed by a fluorescence. If only the singlet fraction of excitons generated in the electroluminescent process is used the limit is 25%, but if the triplets are utilized as well the efficiency can be driven to 100% (here this is the internal efficiency). Effective utilization of triplets was first demonstrated with phosphorescent dopants and later with TADF emitters. The approach explored here is not a new one, but their efficiencies are higher than those reported previously. A group at Kodak reported the use of triplet fusion to enhance OLED efficiencies in 2009. The approach described in the 2009 work is the same one applied here, but the present paper uses better materials and device structures to improve their efficiencies to over 10% (EQE). While this is an advance, it falls well short of the 20% EQEs that both heavy metal based and TADF phosphors have achieved. The paper is not appropriate for Nature Comm on the basis of their device efficiency or basic mechanism for collecting triplets.

Response: We believe the reviewer missed the important point that we are making in this manuscript. In this work, we are not claiming triplet fusion leading to singlet emission in OLEDs is a new phenomenon. Because it takes two triplets to generate a singlet, the device quantum efficiency can never compete with phosphorescent (PHOLEDs) or TADF OLEDs. The important observation we made is light emission is observed at voltages below the bandgap and it appears that the formation of triplets is a direct result of carrier injection without injection of electrons into the frontier LUMO orbital. As a result, if the triplet energy is a half of the singlet energy, the drive voltage of a triplet fusion OLED can be a half of the voltage to drive a PHOLED or TADF OLED. In the OLED community, we generally focus on the quantum efficiency without taking into the account of drive voltage. For applications, it is the power efficiency which determines the power consumption of the device. If the drive voltage is taken into account and the triplet energy is a half of the singlet energy, the power conversion efficiency should be as low as a PHOLED or TADF OLED. In that

case, the power efficiency of our low-voltage triplet fusion OLED should be the same as that of a PHOLED or TADF OLED. That is the point of this article and we believe this is the breakthrough finding of the article which is worthy of publication in Nature Communications.

The authors make the point that their devices have “a power efficiency higher than some of the best phosphorescent blue OLED”. These authors know full well that such a claim is meaningless without comparing spectra. Their EL spectrum has a marked green tail. This impacts the luminance markedly and to compare their devices for their power efficiencies (lm/W) with no mention of spectral purity is misleading. I have not pulled all of the references, but at least some of the devices they compare to are display level blue compounds, making their inherent luminance low do to a poorer match to human photopic response.

Response: Thank you for the very important point. We agree with the reviewer that without considering the spectral information, comparing the device performance is not meaningful. This is in fact the reason why we have compared our device with other devices in the literature that have similar CIE coordinates. It is difficult to find two different emitters with exactly the same emission spectra. An objective way to make a fair comparison of the luminous efficiency is to make sure the two devices with emission spectra having similar CIE chromaticity coordinates. The EL spectrum of BCzVBi has CIE coordinates of (0.15, 0.21), which is deeper than high efficiency phosphorescent blue emitter FIrpic (0.15, 0.32) and most of TADF emitters (*Beilstein J. Org. Chem.* **2018**, *14*, 282). The performance of blue devices listed in table 1 are with similar CIE coordinates, which is a reasonable way to make the comparison. As a result, our comparison with PHOLEDs is fair in the manuscript.

“Moreover, they are comparing their devices, which have low driving voltage, to devices that were not designed to have low voltage. If they had compared their OLEDs to blue phosphorescent OLEDs with conductivity doped HT and ET layers

they would not have seen such a difference. They would have found that the heavy metal phosphor based device outperforms theirs, even before they match spectra.”

Response: Regarding the device operating voltage, there are two important points that we need to differentiate. The first point is the “turn-on voltage” which is related to the charge injection and hence the energetics of the device stack, and is not affected by the conductivity of the charge transport layers. The below bandgap turn-on voltage is the focal point of the present work. The second point is the operating voltage (for example, voltage at operating luminance of 1000 cd/m^2) which is related to the turn-on voltage as well as the conductivity of the charge transport layers. Our low operating voltage is related to the low turn-on voltage of the device. While we agree with the reviewer that conductivity dopants will lower the operating voltage of the device, doping the charge transport layers in our devices will also lower the operating voltage and the authors do not think that it is a fair comparison of the present work with PHOLEDs having doped charge transport layers. For the reviewer’s reference, we summarize the turn-on voltages of p-doped and n-doped blue devices in the following table, along with their spectra. The data below show that our device shows the lowest turn-on. The data below clearly show that doping does not lower the turn-on voltage. Even with p-doping and n-doping, some of the blue phosphorescent devices don’t have a lower operating voltage at 1000 nits than our fluorescent device.

Reference	Emitter	HTL	ETL	Voltage	Efficiency
Our work	BCzVBi	TAPC:Mo O ₃ 10wt%	Bphen	2.5 V @ 1 cd/m ² 3.4 V @ 1000 cd/m ²	14.5 lm/W at 1000 cd/m ²
Organic Electronics 2009, 10, 686	Flr6	MeO-TPD: F4-TCNQ 2 mol%	3TPYMB Cs 20 mol%	3 V @ 0.1 cd/m ²	32 lm/W @ 100 cd/m ² ~26 lm/W @ 1000 cd/m ²
Organic Electronics 2012, 13 2615	Flrpic	TAPC	B3PyPB Liq 25 wt%	2.9 V @ 100 cd/m ²	43.2 lm/W @ 1000 cd/m ²
Applied Physics Letters 2010, 96, 093304	Flrpic	MeOTPD: NDP-2	Bphen:Cs	4.3 V @ 1000 cd/m ²	21 lm/W @ 1000 cd/m ²

There is another issue that the authors seem to have missed or at least chose not to fully explain to the reader. In their device they go to lengths to ensure that the hole/electron recombination generates the triplet directly, without going through the singlet or forming the singlet and triplet in a 1:3 ratio. The triplets then annihilate to form a singlet, which gives rise to emission. The math here is pretty simple. If they are totally useful they will be limited to a theoretical internal efficiency of 50%, and the 10% EQE they see here is a dead-end. They cannot improve beyond this level and report here the best device that will ever be reported.

Response: Maybe the reviewer has missed our point. We never said that the EQE is significantly higher than 10% given the fact that it takes two triplets to generate one

singlet. If the operating voltage can be reduced by 50% which our data may suggest, the power efficiency of these triplet fusion OLEDs can be as high as those for PHOLEDs or TADF OLEDs. At the end, it is the power efficiency, not EQE, that matters in an operating device if one considers the power consumption. Further, an EQE of 10% might not be a “dead-end”. Preferred orientation of the emitter dipoles can further increase the EQE of the resulting device.

The unfortunate thing is that this EQE is well below that of the commercial devices or even average devices. They have chosen to push the power efficiency angle to make their devices more competitive, but as I said above, there are other ways to lower voltage and thus improve power efficiency.

They also make a claim that their devices have low roll-off as they are made brighter. Again, this is a bit of smoke and mirrors. The plot EQE vs. luminance and say that their device is wonderful compared to a CBP based fluorescent device. On a fractional drop bases the two devices are the same, even though one is singlet only and the other is populating exclusively form triplet fusion. Moreover, their plot makes it tough to compare to phosphorscent OLEDs. The normal plot is EQE versus current density. If they replotted their data you would see that the roll-off is pretty typical and that of you look in the literature you will find devices that have much lower roll-off than theirs.

Response: The reviewer said that our EQE is well below that of the commercial devices. This statement is correct when comparing with red and green PHOLEDs. However, it is completely wrong for blue OLEDs! Blue PHOLEDs are not used today in commercial OLEDs because of its poor stability. Instead, blue fluorescent OLEDs are used even with a much lower efficiency compared with the corresponding blue PHOLEDs.

Regarding the efficiency roll-off, the authors strongly disagree with the reviewer that it is a “smoke and mirror”. Anyone who is familiar with OLEDs recognizes that efficiency roll-off is a characteristic of blue PHOLEDs. The roll-off in blue PHOLEDs is due to triplet-triplet annihilation and triplet-polaron quenching

associated with the long triplet lifetime. A typical blue OLED typically has an EQE maximum at about 100 nits and a significant roll-off at 1000 nits. On the other hand, efficiency roll-off in fluorescent OLED is not observed at a luminescence below 1000 nits. For example, the maximum EQE of our present work is at about 3000 nits (corresponding to 10 mA/cm²) whereas the maximum EQE of a typical blue PHOLEDs is at about 200 nits. For the reviewer's reference, we also plot our data in a format of QE vs current density for comparison with a Flrpic PHOLEDs. Here, the maximum EQE is at a current density of 0.1 mA/cm² compared to 10 mA/cm² in our fluorescent device. In the Flrpic device, the EQE decreases by 40% at 10 mA/cm². The efficiency roll-off in phosphorescent OLEDs is obvious.

ITO/MoO₃/TAPC /TCTA:dopant)/TmPyPB/LiF /Al
 Organic Electronics 2013, 14, 316

Based on the previous two paragraphs you would come to the conclusion that I don't think this paper is Nature Communications material. There is a part of this paper that rises to your standard though and may be sufficient to justify publishing it in Nature Communications. This has to do with the direct formation of triplets. I heard Steve Forrest talk about resonant carrier recombination into the triplet exciton some time ago, but I have dug through the literature and have not been able to find any literature report of this. I found a patent, but it has no experimental data, and is not a peer reviewed report in any even. This paper represents the first clear example of being able to directly form the triplet exciton in electroluminescence with high efficiency. The physics of this is very interesting and is discussed in this paper. In my opinion the paper should be built around this process. It will involve a total rework of the paper to shift the emphases away from competing with phosphorescent OLEDs to one of developing new routes to higher efficiency in a device that has the potential to achieve 100% internal efficiency and not one that will dead-end at 50%. If they want to rework their paper along this line I would be happy to rereview it, but in this form, it is not suitable for publication.

Response: We agree with the reviewer that the focus of our work is the device physics rather than device performance. The sub-bandgap turn-on in blue fluorescent OLEDs is very important to the understanding of the operation of an OLED. In the last 20 years, there has been a lot of work on improving the operating lifetime of blue PHOLEDs and no major progress has been made. Whether it is PHOLED or TADF blue OLEDs, it is the high triplet energy causing the degradation. Triplet-triplet annihilation can result in a much higher energy excited states leading to breaking of chemical bonds. That appears to be a "dead end". We strongly disagree that our devices with an internal quantum efficiency of 50% is a dead-end. Yes, the IQE is not as high as a PHOLED. But the voltage can be significantly lower, resulting the same power efficiency with a much lower excitation energy. That is why blue fluorescent OLEDs have a much better lifetime than high efficiency blue PHOLEDs. The importance of our work is two-fold. First, we discovered that OLEDs can be turned on at a voltage corresponding to the triplet energy of the emitter, and we demonstrate the

power efficiency of a fluorescent OLED can be as high as a PHOLED. Second, the implication of our work suggests that the blue OLED lifetime problem can be solved by focusing on triplet fusion OLEDs rather on PHOLEDs or TADF OLEDs. By now, we hope that the reviewer is convinced that our manuscript is worthy of publication in Nature Communications. Since the author cannot provide us the reference for resonant carrier recombination, we cannot comment on that and how it is related to our present work.

Issues for the authors to consider in a revised manuscript:

Comment 1 This is a scientific paper, not a trade journal. The reader should not need to go to the SI to find the structures of the compounds being used here. The first time a compound is referenced in the text the structure should be given. A simple scheme with all of the materials in the paper needs to be added.

Response: Thank you for comment. All materials used in this manuscript are commercial materials and have been widely used in many literatures for a long time. This is not a work about new materials or new material property. The full names and abbreviations are given in the main text and it is not appropriate to give the structures in the main text, and it shouldn't be a problem for readers.

Comment 2 In a revised manuscript I would not see such heavy reliance on the SI. Half of the paper (pages 4-8) rely totally on the SI. This is crazy. The authors should move some of the figures in the SI into the paper. It is not fair to expect the reader to flip back and forth between the paper and the SI so much. The SI is supporting, after all, and if the SI is required to really understand the paper it is no longer "supporting"

Response: Thank you for comment. We have move two figures from SI in the main text, Fig S3 to Fig 2 and S8 to Fig 4.

Comment 3 In the experimental the authors give complete detail on how they do their electrochemistry, but do not talk about how they convert their potentials (relative to

ferrocene) to HOMO energies. There are a number of good papers on this, and they should reference one of them or explain what they did to convert the numbers and why. Their HOMO values are more or less what I expected so I think that this is a matter of the experimental discussion not being complete. There is a much bigger problem with their LUMO values. There is no mention of where these values came from and nearly all of them are wrong, with some off by 1 eV or more. Looking at their values I am guessing that these are optical LUMOs, derived by adding the optical gap to the HOMO energy. The problem is that this is not the LUMO energy. The LUMO energy is estimated electrochemically or determined by IPES. To give you an example, they quote LUMO values for TAPC, CBP and BPhen of -2.0, -2.9 and -3.0, but the correct values are -0.05, -1.8 and -2.0. They miss on both the absolute values and the relative ones as well. They need to explain the origin of their numbers and I suggest they do some measurements or literature digging to get the right values.

Response: Thank you for comment. All of emitter materials used in this manuscript were measured by standard cyclic voltammetry. We have added one sentence to the Experimental Section and an additional reference to clearly articulate how the HOMO levels were determined: “The electrochemical potentials determined from cyclic voltammetry were converted to HOMO values using a modified ionization potential for ferrocene in dichloromethane of 4.89 eV below vacuum.” (Chemistry of Materials 2014, 26 (7), 2414) For further clarification on the determination of the relevant experimental energy levels, we modified the captions in Table S1, Figure S13, and Figure S14 accordingly.

We disagree with this reviewer’s comments on the LUMO values with respect to the vacuum level cited above for TAPC, CBP, and BPhen as our values are correct taken the available literature. As an example, the published LUMO value for BPhen was recently reported by Richard Friend’s group as -3.0 eV (*Adv. Mater.* **2017**, *29*, 1605987), which is exactly the same value we report here. We also list the references of the HOMO and LUMO energy levels of TAPC, CBP in the following table. For every material we cited two references to show that those values are widely accepted

by other researchers.

Materials	HOMO (eV)	LUMO (eV)	Ref. 1	Ref. 2
TAPC	2.0	5.5	Chihaya Adachi Angew. Chem. Int. Ed. 2012, 51, 1-6	Jun Yeob Lee Organic Electronics 2009, 10, 1529
CBP	2.9	6.0	Jang-Joo Kim Thin Solid Films 2013, 531, 419	Jwo-Huei Jou Appl. Phys. Lett. 2010, 96, 143306
BPhen	3.0	6.4	Richard Friend Adv. Mater. 2017, 29, 1605987	Stephen R. Forrest Organic Electronics 2008, 9, 994

Comment 4 Taking the discussion of HOMO and LUMO values above into account, there is a scientific problem they should address. In their single layer devices it is not a problem for eh electrode to deliver electrons at a potential that matches the triplet, but the organic materials are not metals. Their HOMO and LUMO values fixed. That means that in the multilayer device, you can still form the triplet directly, but the voltage benefit should be gone. They clearly show this for a fluorescent device, but why is the same not true for a triplet fusion based device? With their (wrong) LUMO values it looked like you could still inject into the triplet at low voltage, but this should not be the case.

Response: Thank you for comment. In the multi-layer device, the electrons need to overcome the energy barrier from cathode to ETL. Then electrons will direct inject into the triplet energy level of the emitter, BCzVBi, without barrier. In this way, the driving voltage could be slightly higher because of the injection barrier, but the barrier free injection from ETL into EML will reduce the driving voltage.

Comment 5 The authors use the ordinary and extraordinary extinction coefficients to determine the alignment of BCzVBi in a neat thin film. There are two problems with this. The analysis is only useful for single layer devices, when the material is used as a neat thin film. The devices where the alignment issue is important are the ones where

it is present as a dopant. The second problem is that this is not the best method for determining the dopant alignment. Angle dependent PL measurements are far more accurate and have the benefit that they can be used to probe the alignment of a dopant in a film directly. Also, what is the alignment of CzPA? It seems incomplete to only consider a single material in their alignment studies.

Response: Thank you for comment. The purpose of this manuscript is not on dipole orientation. We would like to focus on the triplet formation mechanism and device operation, and decided to remove the ellipsometry data from our manuscript.

Comment 6 The authors continuously refer to the brightness of their OLEDs by the “photocurrent”. I am assuming that they are talking about the photocurrent produced by their photodetector. Since they always give these values as arbitrary units, they should just call it brightness or intensity, not photocurrent. Even better would be to convert the axis to luminance. They do this on some of their plots, why not all of them. Instead of photocurrent change the axes to cd/m^2 .

Response: Thank you for comment. We’ve change the expression “Photocurrent” to “Relative Luminance” for a better understanding and comparison.

Comment 7 Why don’t the CzPA EL spectra in Figures S2 and S3 match?

Response: Thank you for comment. We repeated the measurements and the results are as shown below. The large overlap between the absorption spectrum of BCzVBi and the PL spectrum of CzPA indicates a good energy transfer from the host to the guest. We believe that the inconsistency in our data comes from the batch-to-batch material variation of CzPA we purchased from the vendor. We have repeated the experiments again and correct the PL data are shown below and is now the Fig. S1 in SI. The CzPA EL spectrum is correct in Fig. S4. The peak of CzPA in PL is around 530 nm, which is slightly different from the EL peak at 520 nm. The slight shift in EL is due to the microcavity effect of the CzPA single layer device which we confirmed with optical simulation.

Comment 8 Nature Communications is a general science journal not an optics one. It would be useful to add a sentence or two at line 117 to explain to the reader why the “photocurrent” is quadratic with triplet density at low brightness but changes over to linear at high brightness. Why isn’t it quadratic all the way up? This is a topic that Castellano has discussed at some length in his upconversion work and a couple of sentences here to better explain the physics would be useful for a reader that is new to this area.

Response: Thanks for the useful comment here. We have added several sentences and have revised some of the prior text in the manuscript on page 5 to illustrate the kinetic rationale behind these experimental observations and the associated physics. “it is well established that the up-converted emission intensity displays a quadratic to linear trend as a function of input light excitation intensity between low to high photon flux. These observations are a result of traversing two kinetic limits for the sensitized TF process, the weak-annihilation and strong-annihilation regimes, which only become apparent from the time-integrated expressions. In the weak-annihilation limit, the first order decay (and pseudo-first order quenching by O_2) of the triplet excited state dominates the kinetics resulting in the quadratic power dependence where the fluorescence emission is proportional to the square of the excited triplet concentration. When the TF process dominates, bimolecular triplet annihilation now outcompetes the first-order triplet decay processes and the fluorescence emission becomes directly proportional to the incident light flux exhibiting linear dependence. Precisely the same

phenomenon is to be expected in electroluminescent devices when small-to-large concentrations of annihilating triplets are proportionally generated with increasing driving voltage (which increases current and therefore the triplet concentration), analogous to photochemical upconversion, Kondakov and coworkers have previously shown that in OLEDs exhibiting triplet fusion, at low current density, because of low triplet exciton density, the luminance displays quadratic output behavior vs current density. In this regime the probability of triplets poised to annihilate is low, placing it in the weak annihilation limit. At higher current, the luminance exhibits linear dependence with increasing current density when the TF process becomes dominant.”

REVIEWERS' COMMENTS:

Reviewer #1 (Remarks to the Author):

I would say that the authors have responded to most of my comments and I think the paper should be published after the 3 minor points below are addressed:

- 1) I understand that OLED lifetime measurements are complicated, but I suggest they include the preliminary lifetime measurements in the SI, since they now mention lifetime as a motivating factor in the conclusion. They can emphasize "preliminary" and "future optimization" .
- 2) In the concluding paragraph, they say TTA can reduce "power consumption", but I think it only reduces voltage. You still need twice the current to inject 2 triplets instead of 1 singlet.
- 3) The authors still have not provided a real physical explanation for why triplets are created by direct injection, but that is all right since the phenomenon is quite interesting in itself. But they should at least provide a reference or explanation for why it is unexpected that triplets are formed. This seems to be an unspoken assumption of the field, but there must be a reason why spin singlets are favored. IN either case, the electron is injected into the LUMO, just the spin of its counterpart determines the exciton state.

Reviewer #2 (Remarks to the Author):

The authors have done a reasonable job of addressing my concerns. While I disagree with some of their points, they have added sufficient text to elaborate on the exciting new physics of their approach to make the paper suitable for publications.

One thing I will say is that a respected group publishing a number does not necessarily make it right. The LUMO values in this paper are just wrong. An "optical LUMO" is NOT the transport LUMO (i.e. the real LUMO). Look at your reduction potentials and see if these values you have quoted make any sense for the transport gap. You will see that they don't. This is a big problem in this community. They have accepted that the optical LUMO is the transport LUMO and it is not. It is an ok thing in solar cells, where the optical LUMO is used to see if an excited donor has enough energy to transfer an electron to a given acceptor, but that is not what is going on here. You need the transport LUMO, which can be derived from the electrochemical results or IPES, but not by adding the optical gap to the HOMO energy.

Reviewer #1 (Remarks to the Author):

I would say that the authors have responded to most of my comments and I think the paper should be published after the 3 minor points below are addressed:

1) I understand that OLED lifetime measurements are complicated, but I suggest they include the preliminary lifetime measurements in the SI, since they now mention lifetime as a motivating factor in the conclusion. They can emphasize "preliminary" and "future optimization" .

Response: Preliminary lifetime data are now included in S.I. Figure S12.

2) In the concluding paragraph, they say TTA can reduce "power consumption", but I think it only reduces voltage. You still need twice the current to inject 2 triplets instead of 1 singlet.

Response: The Reviewer is correct. We have rewritten the concluding paragraph to emphasize the point.

3) The authors still have not provided a real physical explanation for why triplets are created by direct injection, but that is all right since the phenomenon is quite interesting in itself. But they should at least provide a reference or explanation for why it is unexpected that triplets are formed. This seems to be an unspoken assumption of the field, but there must be a reason why spin singlets are favored. IN either case, the electron is injected into the LUMO, just the spin of its counterpart determines the exciton state.

Response: We agree with the reviewer on this point and added the following sentences: "Triplet excitons are bound electron-hole pairs with a net spin of one and they are formed when electrons are injected into the LUMO orbital and holes are injected into the HOMO orbital. The conventional wisdom is that they cannot be formed by direct charge injection. On the contrary, our single layer device data suggest that in the presence of holes, the resulting accumulation of holes facilitates injection of electrons leading to direct formation of triplet excitons." We believe that this is an important finding of our work.

Reviewer #2 (Remarks to the Author):

The authors have done a reasonable job of addressing my concerns. While I disagree with some of their points, they have added sufficient text to elaborate on the exciting new physics of their approach to make the paper suitable for publications.

One thing I will say is that a respected group publishing a number does not necessarily make it right. The LUMO values in this paper are just wrong. An "optical LUMO" is NOT the transport LUMO (i.e. the real LUMO). Look at your reduction potentials and see if these values you have

quoted make any sense for the transport gap. You will see that they don't. This is a big problem in this community. They have accepted that the optical LUMO is the transport LUMO and it is not. It is an ok thing in solar cells, where the optical LUMO is used to see if an excited donor has enough energy to transfer an electron to a given acceptor, but that is not what is going on here. You need the transport LUMO, which can be derived from the electrochemical results or IPES, but not by adding the optical gap to the HOMO energy.

Response: We would like to thank the reviewer for making this point. We agree with the reviewer that while it is a common practice to use the optical LUMO level in the literature, the transport LUMO is really the correct value to use, particularly in OLEDs. We have made the following statement to clarify this point in the manuscript: "It should be noted that the LUMO levels used in this work are the literature values which are estimated using optical gap data."